# Polymorphisms and Gene-Gene Interaction in AGER/IL6 Pathway Might Be Associated with Diabetic Ischemic Heart Disease

**DOI:** 10.3390/jpm12030392

**Published:** 2022-03-04

**Authors:** Kuo Liu, Yunyi Xie, Qian Zhao, Wenjuan Peng, Chunyue Guo, Jie Zhang, Ling Zhang

**Affiliations:** 1Beijing Municipal Key Laboratory of Clinical Epidemiology, Department of Epidemiology and Health Statistics, School of Public Health, Capital Medical University, Beijing 100069, China; liukuo@ccmu.edu.cn (K.L.); yiyi95127@126.com (Y.X.); pengwenjuan0311@126.com (W.P.); angela460314@163.com (C.G.); zhangjie@ccmu.edu.cn (J.Z.); 2Department of Biostatistics, FMD K & L, Fort Washington, PA 19034, USA; qiz39@pitt.edu

**Keywords:** diabetic complication, gene-gene interaction, *AGER*, *IL6R*

## Abstract

Background: Although the genetic susceptibility to diabetes and ischemic heart disease (IHD) has been well demonstrated, studies aimed at exploring gene variations associated with diabetic IHD are still limited; Methods: Our study included 204 IHD cases who had been diagnosed with diabetes before the diagnosis of IHD and 882 healthy controls. Logistic regression was used to find the association of candidate SNPs and polygenic risk score (PRS) with diabetic IHD. The diagnostic accuracy was represented with AUC. Generalized multifactor dimensionality reduction (GMDR) was used to illustrate gene-gene interactions; Results: For *IL6R* rs4845625, the CT and TT genotypes were associated with a lower risk of diabetic IHD than the CC genotype (OR = 0.619, *p* = 0.033; OR = 0.542, *p* = 0.025, respectively). Haplotypes in the *AGER* gene (rs184003-rs1035798-rs2070600-rs1800624) and *IL6R* gene (rs7529229-rs4845625-rs4129267-rs7514452-rs4072391) were both significantly associated with diabetic IHD. PRS was associated with the disease (OR = 1.100, *p* = 0.005) after adjusting for covariates, and the AUC were 0.763 (*p* < 0.001). The GMDR analysis suggested that rs184003 and rs4845625 were the best interaction model after permutation testing (*p* = 0.001) with a cross-validation consistency of 10/10; Conclusions: SNPs and haplotypes in the *AGER* and *IL6R* genes and the interaction of rs184003 and rs4845625 were significantly associated with diabetic IHD.

## 1. Introduction

Ischemic heart disease (IHD) remains the leading global cause of death and lost life years in adults, and it is also the leading cause of mortality in people with type 2 diabetes mellitus (T2DM). Approximately 68% of deaths in type 2 diabetic patients are caused by cardiac complications [1,2]. It has been demonstrated that the advanced glycation end products (AGER)/interleukin-6 (IL-6) pathway plays an important role in the physiological mechanism of diabetic cardiovascular complications [3,4,5]; however, whether gene polymorphisms in this pathway can influence the disease susceptibility are still unknown.

Several single nucleotide polymorphisms (SNPs) in the *AGER* gene have been reported to be associated with diabetes or its complications [6,7,8]. A meta-analysis including 27 original articles showed that *AGER* genetic polymorphisms with CAD were potentiated in patients with diabetes mellitus disease, but the association was not consistently significant [9,10,11]. A Mendelian randomization study also showed that the C allele in rs2228145 was associated with a lower risk of coronary heart disease [12], but a meta-analysis of three GWA scans with 4107 type 2 diabetes cases and 5187 controls in Caucasians found no evidence that *IL6R* variants were associated with type 2 diabetes [13]. Due to the different effects of the *IL6R* gene on diabetes and coronary heart disease, further research is still needed to demonstrate the association between *IL6R* variants and diabetic macrovascular complications. Gene-gene interactions can explain the missing heritability of cardiometabolic disease and better reflect the complex pathophysiological process of disease [14]. However, few studies have focused on the influence of the gene-gene interactions among several SNPs on CVD susceptibility until now [15,16,17]. Machine learning methods may effectively reduce Type I and II errors and increase robustness, of which the generalized multifactor dimensionality reduction (GMDR) method has remained popular in detecting the interaction effect since its appearance [18,19].

The current study aimed to illustrate the association of *AGER* and *IL6R* gene polymorphisms with the risk for diabetic ischemic heart disease (IHD) and to assess the modulatory effect of gene-gene interactions between these variants on disease risk. SNPs that were previously reported to be associated with cardiometabolic disease, inflammatory disease, or located in miRNA binding sites were selected for the analysis (see Appendix A).

## 2. Materials and Methods

### 2.1. Study Design and Population

A total of 204 diabetic ischemic heart disease cases and 882 healthy controls were enrolled from communities in Beijing. All subjects gave written informed consent. This study was approved by the Ethics Committee of Capital Medical University (No: 2016SY24).

The inclusion criteria for the cases were as follows: (1) Self-reported or physician-diagnosed diabetes according to the American Diabetes Association Criteria [20]; (2) Ischemic heart disease defined by clinical history, including acute myocardial infarction, angina pectoris, non-ST-elevation acute coronary syndromes, and/or ischemic electrocardiographic alterations; (3) T2DM was diagnosed earlier than ischemic heart disease for at least one year; and (4) Medical records or copies should be provided to verify the diagnosis of diseases.

The exclusion criteria for the cases were as follows: (1) Other diabetic complications: including diabetic nephropathy, diabetic foot, diabetic retinopathy, and diabetic neuropathy; (2) Ischemic cerebrovascular disease or cerebral hemorrhage; (3) Pregnant or lactating women; and (4) Critical physical disability or mental disorder and could not cooperate with the survey.

The inclusion criteria for the controls were as follows: (1) Subjects had not been diagnosed with T2DM before, and fasting blood glucose was less than 5.6 mmol/L in the current survey; (2) Subjects did not have ischemic heart disease, ischemic stroke, or cerebral hemorrhage; (3) Subjects did not have chronic kidney disease; and (4) Subjects were not in the acute phase of infection.

The exclusion criteria for the controls were as follows: (1) Pregnant or lactating women; and (2) Critical physical disability or mental disorder and could not cooperate with the survey.

### 2.2. Measurements

Lifestyle risk factors were obtained from a structured questionnaire. Smoking status was categorized as “currently smoking” and “past/never smoking”. Current smoking was defined as at least 1 cigarette per day, lasting for more than 1 year. Those who had never smoked before or had not smoked for at least 3 months were defined as past/never smoking. Alcohol drinking was categorized as “current alcohol drinking” and “past/never alcohol drinking”. Current drinking was defined as drinking at least once per week and still drinking at that frequency in the previous month. Those who never drank alcohol or had not consumed alcohol for at least one month were defined as never/past alcohol drinking.

Blood pressure (BP) was measured in the morning before participants used antihypertensive medication. Participants were asked to rest for at least 30 min before BP measurement if they had just smoked or had caffeinated products. BP (mmHg) was measured three times at sitting positions by a mercury sphygmomanometer. The average of the last two measurements was used for data analysis.

### 2.3. Serum Markers

After overnight fasting, all participants underwent fasting blood sampling. Fasting blood samples were collected and restored in a 2% EDTA vacutainer for each participant. After centrifugation, the plasma and blood cell samples were separated into two cryovials. Fasting plasma glucose (FPG), total cholesterol (TC), triglycerides (TG), high-density lipoprotein cholesterol (HDLC), and low-density lipoprotein cholesterol (LDLC) were tested using the Beckman Coulter chemistry analyzer AU5800 in the clinical laboratory of Beijing Hepingli Hospital.

Venous blood samples were obtained and stored in a 4 °C refrigerator. All biochemical analyses were performed within 8 h. Serum glucose and biochemical determinations were measured by an enzymatic method using a chemistry analyzer (Beckman LX20, Beckman, Brea, CA, USA) at the central laboratory of the hospital. Highly sensitive C-reactive protein was assessed using a Beckman Coulter chemistry analyzer AU5800 and white blood cell counts were obtained using an AcT5diff cell counter (Beckman Coulter^®^).

### 2.4. Genotyping

Important functional SNPs and previously reported susceptible SNPs were selected as candidate SNPs. Five SNPs (rs1035798, rs1800624, rs1800625, rs184003, and rs2070600) in the *AGER* gene and seven SNPs (rs2228144, rs4072391, rs4129267, rs4537545, rs4845625, rs7514452, and rs7529229) in the *IL6R* gene were selected in the current study.

Genomic DNA was extracted from 1 mL of peripheral blood cells using a TIANGEN DNA kit (TIANGEN Biotech, China, DP319-01) according to the manufacturer’s protocol. The primers were designed by AssayDesigner3.1 software and they were synthesized by Shanghai Thermo Fisher Scientific Co., Ltd., in China. Detailed information on the primers is shown in Appendix A. A Sequenom MassARRAY^®^ matrix-assisted laser desorption/ionization-time of flight mass spectrometry (MALDI-TOF MS) platform (Sequenom Inc., San Diego, CA, USA) was used to genotype SNPs.

### 2.5. Definition of Diseases and Recommendation Level of Their Risk Factors

T2DM was defined as FPG ≥ 7.0 mmol/L or self-reported physician-diagnosed diabetes and/or the use of antidiabetic agents, according to the American Diabetes Association Criteria [21]. Ischemic heart disease (IHD) includes non-fatal acute myocardial infarction [22], angina pectoris [23], acute coronary syndromes [24], and/or ischemic electrocardiographic alterations. Ischemic heart disease in T2DM patients was defined as diabetic ischemic heart disease. Hypertension was defined as systolic blood pressure (SBP) ≥ 140 mmHg and/or diastolic blood pressure (DBP) ≥ 90 mmHg and/or on current antihypertensive medication. Participants with TG ≥ 2.3 mmol/L, TC ≥ 6.2 mmol/L, LDLC ≥ 4.1 mmol/L, or HDLC ≤ 1.0 mmol/L were defined as having dyslipidemia according to the criteria of the 2016 Chinese guidelines for the management of dyslipidemia in adult [25]. The acute phase of infection was defined by hsCRP > 10 mg/L or white blood cell counts > 10.0 × 10^9^. Kidney disease was defined as self-report of diagnosed chronic kidney disease or GFR ≤ 90 mL/min over 3 months. The CKD-EPI equations were used to estimate the glomerular filtration rate [26].

### 2.6. Statistical Analysis

Each continuous variable was tested for normality by the Shapiro-Wilk test. The continuous variables with a normal distribution are expressed as the means ± standard deviations (SD), and the mean difference between groups was tested by Student’s *t*-test. The continuous variables that were non-normally distributed are displayed as the median (interquartile range), and the difference between groups was tested by the Mann-Whitney U test. The categorical variables are expressed as numbers (percentages). The polygenic risk score (PRS) was calculated by summing the number of risk alleles of all candidate SNPs. Logistic regression was used to evaluate the association of diabetic ischemic heart disease with candidate SNPs and PRS, and the diagnostic accuracy was quantified with the area under the ROC curve (AUC). SPSS 25.0 software (SPSS Inc., Chicago, IL, USA) was used for all abovementioned statistical analyses. The generalized multifactor dimensionality reduction (GMDR) method was used to estimate the gene-gene interactions. For the adjustment for multiple testing, a permutation test with 1000 replications was performed. Haplotypes were identified and visualized by Haploview software. The association between haplotypes and diabetic ischemic heart disease and Hardy-Weinberg equilibrium (HWE) was demonstrated by using Plink software (version 1.07). All the SNPs were in HWE (*p* > 0.05). We also performed subgroup analysis by considering FPG and ICVD risk scores. The “10-year ICVD Risk Assessment Form” applicable to the “Cardiovascular Disease Prevention Guidelines in China” was used to estimate ICVD risk score [27]. A two-sided *p* ≤ 0.05 was considered statistically significant.

## 3. Results

### 3.1. General Characteristics of the Studied Participants

A total of 882 healthy controls and 204 diabetic ischemic heart disease cases were included in the current study. The levels of AGEs, TG, FPG, and DBP were significantly higher in the ischemic heart disease cases than in the controls (*p* < 0.001). Serum IL-6 was also higher in the case group, but the difference was not statistically significant. The levels of TC, LDLC, HDLC, and SBP were significantly higher in the controls than in the cases (*p* < 0.001). According to the recommendation of the “2017 Guidelines for the prevention and treatment of type 2 diabetes in China”, the percentages of SBP, DBP, HDLC, LDLC, TG, and TC in the ideal range were significantly higher in the control group than in the case group (*p* < 0.001), see Appendix A. In people with diabetic ischemic heart disease, the proportion of current smokers or alcohol drinkers was significantly lower than in the controls (*p* < 0.001). The details are shown in Table 1.

### 3.2. Association of AGER and IL6R Polymorphisms with Diabetic Ischemic Heart Disease

All polymorphisms were in Hardy-Weinberg equilibrium (all *p*-values were greater than 0.05). For *AGER* rs184003, participants with the GT and TT genotypes had a significantly higher risk of diabetic ischemic heart disease than those with the CC genotype (OR = 1.435, *p* = 0.039; OR = 2.525, *p* = 0.030, respectively). The T allele was associated with an increased risk of diabetic ischemic heart disease by 50% in additive and dominant models (*p* = 0.005; *p* = 0.012, respectively). For *AGER* rs2070600, the T allele was associated with about a 30% lower risk of diabetic ischemic heart disease in the additive and dominant models (*p* = 0.025; *p* = 0.030, respectively). However, after adjusting for potential confounders, the association between the above two SNPs and disease was null. The details are shown in Table 2.

For *IL6R* rs4845625, participants with the CT and TT genotypes had a significantly lower risk of diabetic ischemic heart disease than those with the CC genotype (OR = 0.692, *p* = 0.045; OR = 0.503, *p* = 0.003, respectively). The T allele significantly decreased the risk of diabetic ischemic heart disease in additive and dominant models (OR = 0.707, *p* = 0.003; OR = 0.632, *p* = 0.008, respectively). The association between rs4845625 and disease was still significant after adjusting for potential confounders. The details are shown in Table 2. The association between other SNPs and disease were shown in Appendix A. The polygenic risk score was also associated with an increased risk of diabetic ischemic heart disease by 10% (OR = 1.101, 95% CI: 1.042–1.162, *p* = 0.001). After adjusting for dyslipidemia, hypertension, smoking, and drinking status, PRS was consistently associated with the disease (OR = 1.100, 95% CI: 1.029–1.176, *p* = 0.005).

Compared with models containing only traditional risk factors (AUC = 0.756; 95% CI: 0.714–0.798; *p* < 0.001), the diagnostic accuracy of models containing traditional risk factors together with *IL6R* and *AGER* polymorphisms (AUC = 0.759; 95% CI: 0.716–0.801; *p* < 0.001) and traditional risk factors together with PRS (AUC = 0.763; 95% CI: 0.72–0.80; *p* < 0.001) had slightly higher diagnostic accuracies. However, models containing genetic markers did not improve the diagnostic accuracy significantly (*p* > 0.05). The details were shown in Figure 1.

### 3.3. Association between Haplotypes and Diabetic Ischemic Heart Disease

Four out of five SNPs in the *AGER* gene (Block 1: rs184003-rs1035798-rs2070600-rs1800624) and five out of seven SNPs in the *IL6R* gene (Block 2: rs7529229-rs4845625-rs4129267-rs7514452-rs4072391) showed linkage disequilibrium (see Figure 2). These two blocks were both significantly associated with diabetic ischemic heart disease (Block 1: *p* = 0.008; Block 2: *p* = 0.007). Four haplotypes were constructed in block 1, and two of them were associated with diabetic ischemic heart disease (C-G-T-A: *p* = 0.018; A-G-C-A: *p* = 0.004). Four haplotypes were constructed in block 2, and two of them were associated with diabetic ischemic heart disease (T-C-C-T-C: *p* = 0.033; T-T-C-T-C: *p* = 0.001). The details of the haplotype analysis are shown in Table 3.

### 3.4. The Effect of Gene-Gene Interactions on Diabetic Ischemic Heart Disease

GMDR analysis was performed to assess the effect of gene-gene interactions on diabetic ischemic heart disease risk after adjustment for dyslipidemia, hypertension, smoking, and drinking. The GMDR analysis suggested that rs184003 in the *AGER* gene and rs4845625 in the *IL6R* gene were the best models in terms of statistical significance after permutation testing (*p* = 0.001). The two-locus models had a cross-validation consistency of 10/10 and a testing accuracy of 0.597. Logistic regression was subsequently used to obtain the odds ratios (ORs) and 95% confidence intervals (CIs) for the interaction between rs184003 and rs4845625. In the additive model, the joint effect of rs184003 and rs4845625 was associated with an increased risk of diabetic ischemic heart disease by 38% (OR = 1.38, 95% CI: 1.13–1.69, *p* = 0.002). The GeneMANIA was subsequently used to construct a gene network and predict gene function. As shown in Figure 3, *IL6R* and *AGER* have physical interactions with each other.

### 3.5. Sensitivity Analysis and Subgroup Analysis

The results of sensitivity analysis showed that both *AGER* and *IL6R* polymorphisms were still significantly associated with disease after the adjustment of blood glucose or other potential confounding factors. The interaction between rs4845625 and rs184003 turned to be null after the adjustment of FPG (OR = 1.166, 95% CI: 0.893–1.521, *p* = 0.259). Details were shown in Appendix A.

Subgroup analysis was performed by considering FPG and ICVD risk scores. The results of subgroup analysis showed that rs184003 was significantly associated with disease in higher FPG subgroup (OR = 0.496, 95% CI: 0.288–0.856, *p* = 0.012), and rs4845625 was significantly associated with diabetic IHD in normal ICVD risk score group (OR = 0.389, 95% CI: 0.197–0.768, *p* = 0.007). Details were shown in Appendix A. Subsequently, we found that *AGER* and *IL6R* polymorphisms were not associated with FPG and ICVD risk score, see Appendix A.

## 4. Discussion

Individuals with T2DM have an increased risk of CVD, which cannot be fully explained by elevated glucose [28]. Genetic risk factors contribute greatly to the pathogenesis of diabetic macrovascular complications, but their role has not yet been fully illustrated. In the present community-based case-control study, rs4845625 in the *IL6R* gene and the interaction of rs184003 in the *AGER* gene and rs4845625 in *IL6R* were significantly associated with diabetic ischemic heart disease. The polygenic risk score calculated by summing the number of risk alleles of the SNPs located in the above two genes was also associated with an elevated risk of diabetic ischemic heart disease.

AGER is a multiligand cell surface receptor. Advanced glycation end products (AGEs), which are produced after high glucose exposure, can bind to AGER. Their interaction has been implicated in the pathogenesis of atherosclerosis. In addition, HMGB1 (high-mobility group protein 1) and neutrophil-derived S100 calcium-binding family members (S100A8/A9/A11/A12 and S100B) are ligands of AGER. After ligand binding, pro-inflammatory and pro-coagulant pathways are activated. Rs2070600 was found to be significantly associated with diabetic ischemic heart disease in the current study. However, after adjustments for covariates, the associations became null. Rs2070600 is located in the ligand-binding V domain of the *AGER* gene, often referred to as Gly82Ser [29]. Genome-wide association studies (GWAS) showed that rs2070600 was strongly and dose-dependently correlated with sRAGE levels in whites and blacks from the Atherosclerosis Risk in Communities Study and a Chinese population [30,31]. Interestingly, although soluble RAGE levels were found to be associated with diabetic complications in many studies, the association between rs2070600 and ischemic heart disease or other diabetic complications was not consistent. In the Atherosclerosis Risk in Communities Study, rs2070600 was not significantly associated with incident coronary heart disease or diabetes in either whites or blacks with a median follow-up of 20 years [30]. Chinese researchers [32] found a significant association between rs2070600 and coronary arterial disease in 175 cases and 170 controls. A meta-analysis found that the discrepancy may be attributable to ethnicity, and subjects with the rs2070600 risk allele were at higher risk of coronary arterial disease (CAD) in the Chinese population than in the non-Chinese population [11]. However, our study found that the association between rs2070600 and diabetic ischemic heart disease was null. Another study also found that rs2070600 was associated with the circulating levels of esRAGE but not with CAD in Chinese patients with T2DM [33]. These results might indicate that the association between rs2070600 and CAD may also be different in the general population and T2DM patients.

Only a few studies have demonstrated the association between rs184003 and ischemic heart disease. In the current study, we also found that the haplotypes C-G-T-A and A-G-C-A in the *AGER* gene (rs184003-rs1035798-rs2070600-rs1800624) were significantly associated with diabetic ischemic heart disease. Additionally, the association between haplotypes and potential confounders were null (Appendix A). A hospital-based case-control study involving 1142 patients diagnosed with CAD and 1106 age- and sex-matched controls in a Chinese population was better powered and designed. In this study, the T allele in rs184003 and haplotypes in the *AGER* gene (rs1800625-rs1800624-rs2070600-rs184003, C-T-G-G and T-A-G-T) were also found to be associated with an increased risk of CAD [34]. In addition to the significant association with CAD, the result of a meta-analysis also showed that the homogeneity of the rs184003 polymorphism with the T allele conferred an increased risk of diabetes mellitus in East Asians (OR = 1.21; 95% CI: 1.04–1.40; I^2^ = 0) [35]. A previous study conducted in a Chinese population involving 200 gastric cancer patients and 207 cancer-free controls showed that subjects carrying the rs184003 T variant allele had an increased ability to produce soluble RAGE (sRAGE) [36]. Given that the T allele in rs184003 was associated with a higher risk of both diabetes and CAD, sRAGE might act on the common pathogenetic pathways of cardiovascular and metabolic diseases. Soluble RAGE levels were found to be significantly associated with CAD and diabetes [37,38,39] in many studies, and the association between haplotypes in the *AGER* gene and diabetic ischemic heart disease in the current study indicated that sRAGE levels could serve as a marker of diabetic ischemic disease. A recent review demonstrated that RAGE signaling contributed to vascular calcification in diabetic and nondiabetic subjects, presumably on account of the generation of RAGE ligands such as AGEs and other proinflammatory/pro-oxidative ligands [40]. The current study also found that the level of AGEs was elevated in the diabetic ischemic disease group, which supports the above hypothesis. To our knowledge, few studies have illustrated the relationship between rs184003 and diabetic macrovascular complications. Although the findings suggest potential benefits with RAGE antagonism both in the causes and consequences of diabetes and its macrovascular complications, more research is still needed to validate our results.

Mendelian randomization analysis illustrated that IL6R signaling might have a causal role in the development of coronary heart disease [12]. A previous meta-analysis demonstrated that the C allele in rs7529229 *IL6R* was associated with a lower risk of coronary heart disease [12,41]. Although the meta-analysis included a large sample size and better designed original studies, the populations of the studies were all Caucasian, so the evidence from Asian populations was still insufficient. In the current study, the association between rs7529229 and diabetic ischemic heart disease was null in the Chinese population. Chen et al. also did not find a significant effect of rs7529229 on coronary stenosis or acute myocardial infarction in the Chinese Han population with a sample size of 187 patients and 231 controls [42]. However, according to the sample size, allele frequency, and OR reported in the above study, the statistical power was relatively low and might lead to false negative results. Thus, studies with larger sample sizes are still needed to replicate the above findings. Likewise, He et al. conducted a hospital-based case-only study in 402 patients with left main coronary artery disease (LMCAD) and 804 patients with more peripheral coronary artery disease (MPCAD) in a Chinese population, and the results showed that rs7529229 CC or TC/CC genotypes were associated with an increased risk of LMCAD compared with MPCAD [43]. The haplotype T-T-C-T-C (rs7529229-rs4845625-rs4129267-rs7514452-rs4072391) in the *IL6R* gene and rs4845625 were associated with diabetic ischemic heart disease in our study, and the association held after adjusting for potential confounders. In addition, haplotypes T-T-C-C-T were significantly associated with TC (Appendix A). Rs4845625 was found to be significantly associated with hypertriglyceridemia in the Japanese population [44], and the T allele was associated with a lower serum concentration of creatinine and increased EGFR [45]. Hypertriglyceridemia and chronic kidney disease (CKD) have common pathways, such as endothelial dysfunction, dyslipidemia, and inflammation, leading to metabolic cardiovascular disease [20]. Although few studies have focused on the association between rs4845625 and diabetic heart disease, its association with triglycerides and kidney function might indicate the potential mechanisms of rs4645625 in diabetic ischemic heart disease.

In response to hyperglycemia, AGER is activated by S100A8/A9 on hepatic Kupffer cells, leading to the secretion of IL-6. IL-6 subsequently binds to its receptor (IL6R) on hepatocytes to enhance the production of thrombopoietin, thereby regulating platelet production and resulting in diabetes-induced thrombocytosis [46]. In the current study, we found that the gene-gene interactions between *AGER* and *IL6R* increased the risk of diabetic ischemic heart disease. We subsequently used GeneMANIA to construct a gene network and predict gene function. *IL6R* and *AGER* have physical interactions with each other, and several pathways, including NF-kB/RelA and JAK/STAT, are involved in these interactions (Figure 3). The function of *AGER* and *IL6R* polymorphisms were listed in Appendix A. These interactions illustrated that the interaction of SNPs in *IL6R* and *AGER* was not only a statistical interaction but also a biological interaction. To our knowledge, this is the first study aimed at identifying the interaction of the *AGER* and *IL6R* genes, and our results provide genetic evidence on the physiological mechanism of diabetic macrovascular complications. Whether the main effect and gene-gene interactions in these two genes could be used to predict the risk of diabetic macrovascular complications still needs to be validated by cohort studies in the future. Although we found a significant interaction between the *AGER* gene and the *IL6R* gene, the association between circulating IL-6 and diabetic ischemic heart disease was null. This result indicated that *IL6R* polymorphisms still need to be further demonstrated. The most common hypothesis is that IL-6 in hematopoietic cells, but not circulating IL-6, were more likely to affect TPO production and macrovascular complications [46,47].

The results of the sensitivity analysis showed that both *AGER* and *IL6R* polymorphisms were still significantly associated with disease after the adjustment of different risk factors, respectively. In the current study, SBP, TC, and LDLC levels and the proportion of people with smoking and drinking habits were significantly lower in the cases than in the controls, which is not consistent with other studies. According to the “2017 Guidelines for the prevention and treatment of type 2 diabetes in China”, diabetes patients have more stringent standards on blood pressure (BP) and blood lipids than the healthy population, and diabetes patients with ischemic heart disease should quit smoking and drinking [48]. Diabetes patients might change their lifestyles and medication to maintain their BP or blood lipids at a lower level. Due to the case-control study design of the current study, we were not able to collect lifestyle risk factors and blood samples before the incidence of diabetic ischemic heart disease. However, the percentages of SBP, DBP, HDLC, LDLC, TG, and TC in the ideal range were significantly higher in the control group than in the case group (*p* < 0.001, Appendix A). Due to the above limitation of our study, more longitudinal studies are still needed to demonstrate whether genetic variants will increase the incidence of diabetic macrovascular complications. In addition, we did not collect any information about diabetic ketoacidosis in the current study. Since diabetic ketoacidosis predisposes individuals to ischemic heart disease, our study might induce bias to a certain extent. Moreover, medication information was not included in the investigation. Given that some antidiabetic medications, such as SGLT-2 inhibitors [49], will reduce the risk of ischemic heart disease in diabetes patients, future studies considering antidiabetic medication are still needed to validate the genetic effect on diabetic macrovascular complications. Since we did not recruit participants who only have diabetes or only have ICH, we performed subgroup analysis by considering FPG and ICVD risk scores. The ICVD risk scores were simple used as a tool to reflect the cumulative risk factors of ICVD in order to rule out the possibility of candidate SNPs that affect IHD. The subgroup analysis helps to rule out the possibility that diabetes or the ICVD risk score are confounders of the current study. Future research, including T2DM group without complications and ischemic heart disease (IHD) group without diabetes, would provide additional information in clarifying the roles of the *AGER* gene and *IL6R* gene.

## 5. Conclusions

Haplotypes in the *AGER* gene (C-G-T-A and A-G-C-A) were risk factors for diabetic ischemic disease, and rs4845625 and haplotypes in the *IL6R* gene (T allele and T-T-C-T-C) were associated with a lower risk of diabetic ischemic heart disease. The gene-gene interactions between rs184003 in *AGER* and rs4845625 in *IL6R* were associated with a higher risk of diabetic ischemic heart disease.

## Figures and Tables

**Figure 1 jpm-12-00392-f001:**
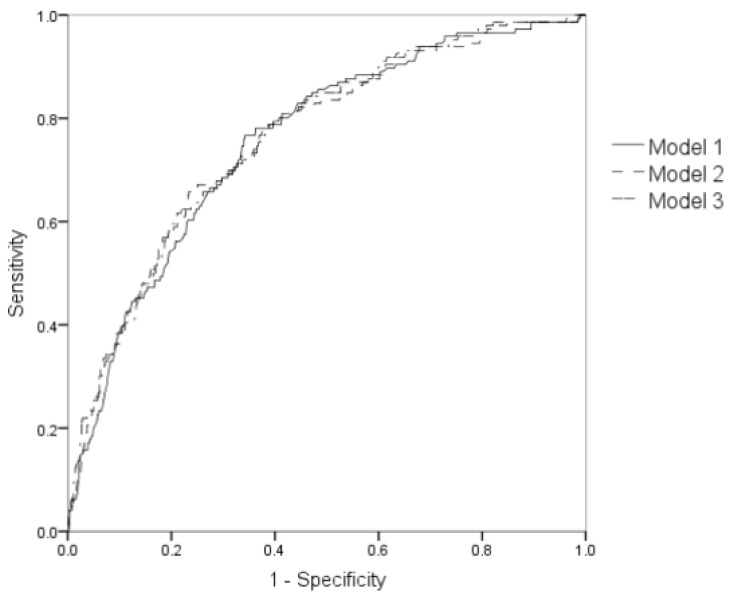
ROC curve of different statistical models. Models were built by logistic regression, variables contained in each model were as follows: Model 1: Age, sex, hyperlipidaemia, hypertension, smoking, and alcohol drinking behavior. Model 2: Variables in model 1 and rs184003 and rs4845625. Model 3: Variables in model 1 and PRS.

**Figure 2 jpm-12-00392-f002:**
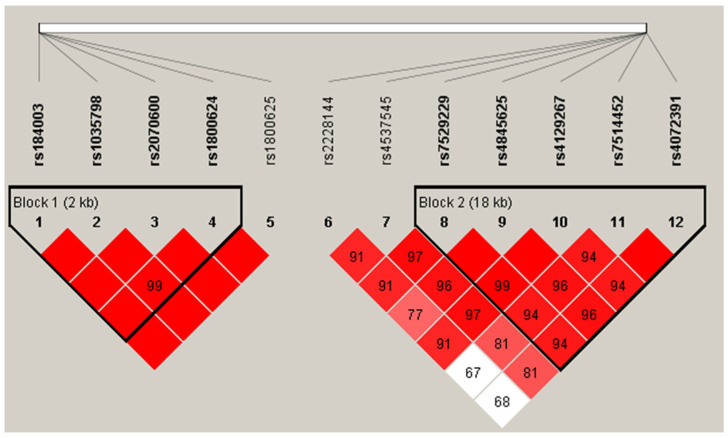
Haplotypes in *AGER* gene and *IL6R* gene. Two haplotypes were identified by haploview software. *AGER* gene: rs184003-rs1035798-rs2070600-rs180062; *IL6R* gene: rs7529229-rs4845625-rs4129267-rs7514452-rs4072391.

**Figure 3 jpm-12-00392-f003:**
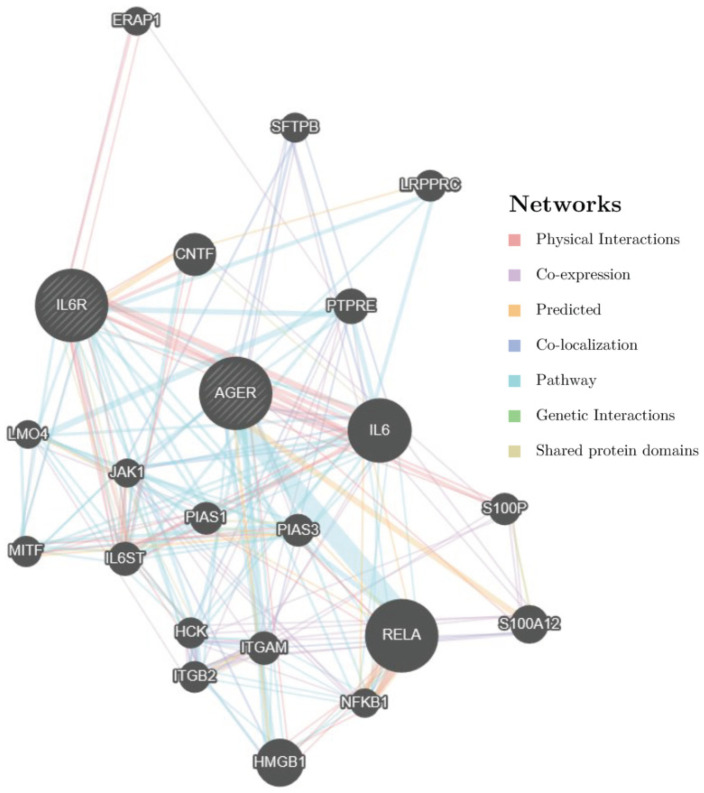
The gene network between *AGER* and *IL6R*. According to the gene network constructed by GeneMANIA, *IL6R* and *AGER* have physical interactions with each other.

**Table 1 jpm-12-00392-t001:** Demographic and biochemical characteristics of the participants.

	Controls	T2DM + IHD	*p* Value
Age (years) ^1^	64.00 ± 11.25	65 ± 11.00	0.126
Male (n, %)	488 (55.3)	119 (58.3)	0.436
SBP (mmHg) ^1^	136.00 (24.00)	130.00 (19.50)	<0.001 **
DBP (mmHg) ^1^	79.00 (14.00)	80 (12.00)	0.001 **
BMI (kg/m^2^) ^1^	25.64 (4.46)	25.36 (3.53)	0.578
TC (mmol/L) ^1^	5.12 (1.41)	4.68 (1.53)	<0.001 **
HDLC (mmol/L) ^1^	1.34 (0.48)	1.22 (0.45)	<0.001 **
LDLC (mmol/L) ^1^	3.02 (1.17)	2.68 (1.04)	<0.001 **
TG (mmol/L) ^1^	1.40 (0.90)	2.35 (1.57)	<0.001 **
FPG (mmol/L) ^1^	5.60 (0.93)	7.07 (3.53)	<0.001 **
Current smoking (n, %)	194 (22.0)	18 (8.8)	<0.001 **
Current drinking (n, %)	295 (33.5)	32 (15.7)	<0.001 **
AGEs (mmol/L) ^1^	31.05 (15.94)	38.07 (16.82)	<0.001 **
IL-6 (mmol/L) ^1^	133.08 (49.32)	136.83 (40.18)	0.353

^1^ Variables, which have non-normal distribution, were displayed as median (interquartile range), and were tested by Mann-Whitney U test. T2DM: type 2 diabetes, IHD: ischemic heart disease, BMI: body mass index, SBP: systolic blood pressure, DBP: diastolic blood pressure, FPG: fasting plasma glucose, TG: triglyceride, TC: total cholesterol, LDL-C: low density lipoprotein cholesterol, HDL-C: high density lipoprotein cholesterol, AGEs: advanced glycation end products, IL-6: interleukin 6. ** *p* < 0.01.

**Table 2 jpm-12-00392-t002:** Associations of rs184003, rs2070600, and rs4845625 with the risk of diabetic cardiovascular disease.

	Genotype	Crude OR ^1^ (95%CI)	Crude *p* Value	Adjusted OR¤ (95%CI)	Adjusted *p* Value
rs184003	GG	Ref	Ref	Ref	Ref
	GT	1.435 (1.019, 2.020)	0.039 *	1.223 (0.797, 1.878)	0.357
	TT	2.525 (1.092, 5.837)	0.030 *	1.651 (0.580, 4.702)	0.348
	additive	1.491 (1.125, 1.976)	0.005 **	1.247 (0.880, 1.767)	0.215
	dominant	1.518 (1.093, 2.017)	0.012 *	1.265 (0.839, 1.905)	0.261
	recessive	2.282 (0.993, 5.241)	0.046	1.571 (0.555, 4.449)	0.395
rs2070600	CC	Ref	Ref	Ref	Ref
	CT	0.713 (0.496, 1.024)	0.067	0.843 (0.550, 1.294)	0.435
	TT	0.536 (0.237, 1.211)	0.134	0.611 (0.206, 1.807)	0.373
	additive	0.721 (0.542, 0.960)	0.025 *	0.819 (0.578, 1.162)	0.264
	dominant	0.684 (0.485, 0.964)	0.030 *	1.399 (0.914, 2.140)	0.122
	recessive	0.587 (0.261, 1.320)	0.198	2.204 (0.493, 9.851)	0.301
rs4845625	CC	Ref	Ref	Ref	Ref
	CT	0.692 (0.483, 0.991)	0.045 *	0.619 (0.398, 0.961)	0.033
	TT	0.503 (0.318, 0.795)	0.003 **	0.542 (0.318, 0.924)	0.025
	additive	0.707 (0.563, 0.888)	0.003 **	0.732 (0.557, 0.961)	0.025
	dominant	0.632 (0.448, 0.889)	0.008 **	0.594 (0.392, 0.902)	0.014
	recessive	0.644 (0.434, 0.955)	0.028	0.757 (0.481, 1.191)	0.229

^1^ No variables were adjusted in logistic regression model ¤ Dyslipidemia, hypertension, smoking, and drinking were adjusted in the logistic regression model. Adjusted *p*-values shown in the table are adjusted only by covariates. * *p* < 0.05, ** *p* < 0.01.

**Table 3 jpm-12-00392-t003:** Haplotype analysis for blocks in *AGER* and *IL6R* genes.

	Haplotypes	F_U ^1^	F_A¤	Chi-Square	OR (95%CI)	*p* Value
Block 1 ^2^	Omnibus test	-	-	11.750		0.008 **
	C-A-C-T	0.162	0.170	0.162	1.049 (0.830, 1.327)	0.687
	C-G-T-A	0.202	0.150	5.575	0.743 (0.580, 0.951)	0.018 *
	A-G-C-A	0.140	0.197	8.229	1.407 (1.114, 1.777)	0.004 **
	C-G-C-A	0.497	0.482	0.247	0.970 (0.859, 1.094)	0.620
Block 2 ^3^	Omnibus test	-	-	11.99		0.007 **
	T-T-C-C-T	0.093	0.100	0.227	1.075 (0.798, 1.449)	0.634
	C-C-T-T-C	0.387	0.431	2.639	1.114 (0.978, 1.268)	0.104
	T-C-C-T-C	0.095	0.131	4.551	1.379 (1.026, 1.853)	0.033 *
	T-T-C-T-C	0.426	0.338	10.32	0.793 (0.689, 0.914)	0.001 **

^1^ F_U: minor allele frequency in controls; ¤ F_A: minor allele frequency in cases; ^2^ Block1: rs184003-rs1035798-rs2070600-rs1800624; ^3^ Block2: rs7529229-rs4845625-rs4129267-rs7514452-rs4072391; * *p* < 0.05; ** *p* < 0.01.

## Data Availability

Research could contact communication author to access data.

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
