# Peer review of "Polymorphisms and Gene-Gene Interaction in AGER/IL6 Pathway Might Be Associated with Diabetic Ischemic Heart Disease"

_jpm, 2022, doi:10.3390/jpm12030392_

Round 1
Reviewer 1 Report
The authors concluded that single nucleotide polymorphisms(SNPs) and haplotypes in the AGER and IL6R genes and the interaction of rs184003 and rs4845625 were significantly associated with diabetic IHD.
The abstract may draw attention of the clinicians because this research seems to reveal the mysterious relationship between diabetes and ischemic heart disease from a new aspect of genes.
This research has some limitations as the authors admitted. First, they "didn't recruit participants who only have diabetes or only have ICH (probably a typo of IHD)" (line 447-448). They should have compared diabetes only patients and healthy controls, as well as IHD only patients and healthy controls to prove their conclusion. Otherwise, we cannot tell if the difference was caused by diabetes, IHD, or both.
Second, the sample size is not balanced. There were 882 healthy controls to compare with only 204 cases.
Third, the control group had higher IHD risk factors.
The idea of this research is novel and the observation is very impressive. However, the data is too big and not very organized. Please make it more simple and easier for the readers.
As for the units, please use mg/L instead of mmol/L.
Reviewer 2 Report
Liu et al. performed an interesting study regarding the polymorphisms and interactions implicated in diabetic ishecmic heart disease. I habe the following comments: Introduction is way too long for such article. The authors should abbreviate it and concentrate on presenting its aim. Diabetic ishemic heart disease should be defined more clearly. How much time before IHD indices occurrence in relation to onset of diabetes should have passed before it was defined as diabetic? It is not clear what implications can these onservation have. The authors should expand the discussion section with this but also with future directions arising from these conclusions. Perhaps the biggest setback is unknown relation to therapy of diabetes, and thus the value of the obtained results is questionable.Author Response
Please see the attachment.
Reviewer 3 Report
To the editor: Dear editor, in my opinion, the article is ready to publish after minor corrections and inclusion of a figure showing mechanism of action.
To authors: Overall, the manuscript entitled “Polymorphisms and gene-gene interaction in AGER/IL6 path- way are associated with diabetic ischemic heart disease” is a very well documented article on new risk factors found in association with diabetic ischemic disease.
Specific comments:
- Abstract: Good concision and clarity.
- Introduction: this chapter has enough information about the subject of the study.
- Materials and Methods: very comprenhensive information. In statistical analysis section, where plink software is mentioned, please include software version.
- Results: in general characteristics of the studied participants section, the paragraph "According to the recommendation of the “2017 Guidelines for the prevention and treat- ment of type 2 diabetes in China”, the percentages of SBP, DBP, HDLC, LDLC, TG and TC in the ideal range were significantly higher in the control group than in the case group (P<0.001)" needs be supported by data. Please add data referred to this paragraph.
- Discussion: This chapter is well written and comparison with other published works is well balanced.
- Conclusions: well explained and concise chapter.
- Tables: tables and Ssupplementary data are very appropiate.
- Figures: Figures are very illustrative. In opinion of this reviewer, figure A1 from supplementary data should be included in discussion section of the main manuscript.
Round 2
Reviewer 1 Report
My first comment in the previous review :
First, they "didn't recruit participants who only have diabetes or only have ICH (probably a typo of IHD)" (line 447-448). They should have compared diabetes only patients and healthy controls, as well as IHD only patients and healthy controls to prove their conclusion. Otherwise, we cannot tell if the difference was caused by diabetes, IHD, or both.
The authors wrote lengthy reply to make up this limitation. As the authors admitted, this is a critical limitation of this research and the other readers will have the same impression as I do. Please reconsider the study design.
The introduction became shorter, but the line 58 "This result provides evidence on the precise prevention of IHD in diabetes" is too strong and too exaggerated. This research shows interesting observation, but it is difficult to apply to clinical daily practice. Gene evaluation is costly and not practical for office practice. The line 387, "Our results provided evidence that genetic polymorphism can be used to identify the susceptible population of diabetic IHD" is too strong. Gene testing is too expensive to identify the susceptible population, thus cannot be done for whole nation.
The authors wrote in line 378-379, "the association between circulating IL-6 and diabetic ischemic heart disease was null." Therefore, all the readers will get confused ; then what is the purpose of this research ? And why the authors wrote in the line 393, "Our results implicated that AGER/IL6R pathway contribute to diabetic IHD" ? The line 394-395 is also too strong or exaggerated because this research is not about IL-6 associated diseases.
Reviewer 2 Report
Unfortunately, I see no attached file for my comments.
Round 3
Reviewer 1 Report
My first comment of the 2nd review and the reply from the authors:
- of IHD)" (line 447-448). They should have compared diabetes only patients and healthy controls, as well as IHD only patients and healthy controls to prove their conclusion. They "didn't recruit participants who only have diabetes or only have ICH (probably a typo Otherwise, we cannot tell if the difference was caused by diabetes, IHD, or both. The authors wrote lengthy reply to make up this limitation. As the authors admitted, this is a critical limitation of this research and the other readers will have the same impression as I do. Please reconsider the study design.
Reply: Thank you for your comment. We felt sorry about that, but to the ethnic concern, we cannot add comparison groups at this time. We will definitely make up this limitation in the future research. In the present article, we have clearly pointed out the case and control groups in the abstract. Readers can judge by themselves through the abstract, and there are also subgroup analysis and sensitivity analysis in the full text to make up for this limitation.
In my opinion, this is a critical failure of study design. What is the rush of publication in the current form ? Please reconsider to add comparison groups to make the research more scientifically sound.
My 4th comment of the previous review and the authors' reply:
The authors wrote in line 378-379, "the association between circulating IL-6 and diabetic ischemic heart disease was null." Therefore, all the readers will get confused; then what is the purpose of this research?
Reply: Thank you for comment. The serological levels of IL-6 receptor and IL-6 are not necessarily identical. IL-6 signals via glycoprotein 130 (gp130) and the membrane-bound or soluble IL-6 receptor (IL-6R), referred to as classic or trans-signaling, respectively. Soluble IL-6R (sIL-6R) and sgp130 constitute a buffer system and increases the serum half-life of IL-6 or restricts systemic IL-6 signaling. Therefore, we found that IL-6R-related gene polymorphism was associated with disease, but IL-6 level was not associated with disease, indicating that the biological function of IL-6 is complex. See line 378-383.
The explanation about the relationship among IL-6, IL-6R, and IHD is vague and complicated in this report. The authors are also giving wrong impression to the readers that they found the key to solve the mystery why tocilizumab reduces the risk factors of IHD (line 386-396). As IL-6 or IL-6R is not regarded as the risk factors of IHD, the authors had better not mention much about it.
Author Response
1. Thank you for your comment. As we mentioned before, the comparison group could not recruit due to the ethical concern at the current moment. In considering the current design cannot strongly confirm the association with diabetic ischemic heart disease, we revised the title of paper from “are associated” to “might be associated” as follow: “Polymorphisms and gene-gene interaction in AGER/IL6 path-way might be associated with diabetic ischemic heart disease.”
2. Thank you for comment. The serological levels of IL-6 receptor and circulating IL-6 are not necessarily identical. Thus, the null association between circulating IL-6 and diabetic ischemic heart disease did not indicate IL6R and its genetic polymorphism had no effect on the disease. Many gene polymorphisms have been found to be associated with diseases, but their functions and mechanisms have not been clarified. Therefore, the null association between circulating serum marker and disease would not deny our main results. In order to better describe our opinion, we revised our description as following: “Although we found a significant interaction between the AGER gene and the IL6R gene, the association between circulating IL-6 and diabetic ischemic heart disease was null. This result indicated that IL6R polymorphism are still need to be further demonstrated. The most common hypothesis is that IL-6 in hematopoietic cells, but not circulating IL-6, were more likely to affect TPO production and macrovascular complications [46,47].”
3. Thank you for comment. We have deleted the content about tocilizumab as you mentioned (line 386-396).
Reviewer 2 Report
The authors properly addressed my concerns
Author Response
Thank you.